# Hair Cortisol Concentration, Perceived Stress, Mental Well-Being, and Cardiovascular Health in African American Older Adults: A Pilot Study

**DOI:** 10.3390/geriatrics7030053

**Published:** 2022-04-29

**Authors:** Ericka L. Richards, Kathy D. Wright, Ingrid K. Richards Adams, Maryanna D. Klatt, Todd B. Monroe, Christopher M. Nguyen, Karen M. Rose

**Affiliations:** 1Center for Healthy Aging, Self-Management, and Complex Care, College of Nursing, The Ohio State University, Columbus, OH 43210, USA; richards.1253@osu.edu (E.L.R.); monroe.1181@osu.edu (T.B.M.); rose.1482@osu.edu (K.M.R.); 2School of Health & Rehabilitation Sciences, Medical Dietetics, The Ohio State University College of Medicine, Columbus, OH 43210, USA; richards.3@osu.edu; 3Department of Extension, College of Food, Agricultural, and Environmental Sciences, The Ohio State University, Columbus, OH 43210, USA; 4Center for Integrative Health, Department of Family and Community Medicine, The Ohio State University College of Medicine, Columbus, OH 43201, USA; klatt.8@osu.edu; 5Department of Psychiatry & Behavioral Health, The Ohio State University College of Medicine, Columbus, OH 43214, USA; christopher.nguyen@osumc.edu

**Keywords:** hair cortisol concentration, blood pressure, hypertension, stress, African American, older adults, mental well-being, mean arterial pressure, cardiovascular health, SF-36 mental components summary, diastolic, systolic

## Abstract

(1) Background: African Americans experience high rates of psychological stress and hypertension, which increases their risk of cardiovascular disease with age. Easy-to-collect psychological and biological stress data are valuable to investigations of this association. Hair cortisol concentration (HCC), as a proxy biomarker of chronic stress exposure, provides such advantages in contrast to collection of multiple daily samples of saliva. Objective: To examine the relationships among HCC, perceived stress, mental well-being, and cardiovascular health (systolic blood pressure (SBP), diastolic blood pressure (DBP), and mean arterial pressure (MAP)). (2) Methods: Cross-sectional secondary data (*N* = 25) were used from a mind–body intervention study in hypertensive African Americans ages 65 and older. Data included HCC, a four-item perceived stress scale, SF-36 mental components summary, and SBP/DBP. SBP + 2 (DBP)/3 was used to calculate MAP. (3) Results: The relationship between mental well-being and perceived stress (r = −0.497, *p* ≤ 0.01) and mental well-being and DBP (r = −0.458, *p* = 0.02) were significant. HCC change was not significant. In a regression model, every unit increase in well-being predicted a 0.42 decrease in DBP (β = −0.42, 95% CI (−0.69–0.15)) and a 1.10 unit decrease in MAP (β = −1.10, 95% CI (−1.99–0.20)). (4) Conclusions: This study contributes to the knowledge of physiologic data regarding the relationship between MAP and well-being. Findings from this study may aid in the development of interventions that address mental well-being and cardiovascular health in African American older adults with hypertension.

## 1. Introduction

Hypertension is a significant risk factor for cardiovascular disease [1], and according to a longitudinal study examining the causes of heart disease in African Americans, perceived stress over time predicts incident hypertension (systolic blood pressure 130–139 mm Hg or diastolic blood pressure 80–89 mm Hg), adjusting for other hypertension risk factors, such as genetics and lifestyle [2]. Older African Americans experience 2.7 times greater stress than whites of similar age [3]. This inequality reflects socioeconomic disadvantage including low education, dwelling in poor neighborhoods, structural racism, and discrimination [2,4,5]. Likewise longitudinal, nationally representative data shows that African Americans report higher stress levels and have greater exposure to chronic stressors in their lifetime than whites [3]. However, such research is based on subjective surveys and other research relying on cumbersome measures of cortisol, such as saliva, urine, or blood, that require multiple samples to ascertain chronic stress exposure [6,7,8]. Thus, our understanding of chronic physiologic stress is incomplete, and prioritizing the examination of psychological stress using less burdensome methods is imperative to mitigate the complications of hypertension and improve cardiovascular health in African Americans.

Cortisol from scalp hair provides a measurement of hypothalamic–pituitary–adrenal axis (HPA) axis activity over the preceding months, much like hemoglobin A1C is a proxy measure of glucose control over time [6,7,8]. Thus, scalp hair cortisol concentration (HCC) is a proxy biomarker for chronic stress because it measures HPA function and exposure to cortisol elevation over time [6,7,8]. Indeed, research demonstrates associations between HCC and diabetes [9], metabolic syndrome [10], and resilience [10]. Gaps remain in the use of HCC in African Americans 65 and older as a biologic measure of long-term stress exposures and HCC’s association with mental well-being, perceived stress, and cardiovascular health (systolic blood pressure (SBP), diastolic blood pressure (DBP), and mean arterial pressure (MAP)).

Increased MAP, an indicator of arterial stiffness and risk factor for cardiovascular disease, has been associated with higher HCC in middle-aged women [10], but to our knowledge, there are no studies that have examined this relationship in African American older adults. In addition, studies that use MAP tend to (1) focus on individuals with diagnoses of pulmonary hypertension or congestive heart failure, (2) have a limited number or no African Americans in their sample, and (3) report physical function and not emotional well-being [11,12,13]. As the African American population ages, it is imperative to examine factors that contribute to poor cardiovascular health.

Our pilot study adds to existing knowledge by (1) engaging a traditionally unheard group, African American older adults, in psychological stress and HCC biomarker research; (2) outlining a clear procedural methodology that other scientists can replicate; and (3) examining modifiable psychological risk factors (perceived stress and emotional well-being) in the context of a biomarker for chronic stress exposure, HCC. The secondary analysis presented here examines the relationship between HCC, mental well-being, perceived stress, and cardiovascular health (systolic/diastolic blood pressure and MAP) in African American older adults. We hypothesized that HCC would have positive correlations with both perceived stress and cardiovascular health (systolic/diastolic and MAP) and an inverse correlation with mental well-being. In other words, we expect lower cortisol to correlate with lower perceived stress, better cardiovascular health, and higher well-being.

## 2. Materials and Methods

### 2.1. Study Design and Participants

This cross-sectional pilot study was a secondary analysis of baseline data from a mind–body intervention for African American older adults that was collected from August 2018 to August 2019 [13]. The parent study was a randomized, controlled trial of Mindfulness in Motion plus Dietary Approaches to Stop Hypertension intervention to improve diet, increase mindfulness, and reduce stress and systolic blood pressure that used an attention control (educational lectures) or true control group [13]. Study criteria included self-identity as African American or Black, 65 years of age and older, hypertension, and able to provide consent. Exclusion criteria were the use of glucocorticoids six months prior to the study, Cushing’s disease, end-stage chronic kidney disease requiring dialysis, and severe cognitive impairment. A subset was used that included those participants who provided a hair sample. Participants who did not have enough hair to provide a sample or who refused to provide a sample were excluded from the subset. The Ohio State University Institutional Review Board approved the study, and written informed consent was obtained from participants. The Research Electronic Data Capture (REDCap) system was used for data collection throughout the study, and data were stored under the participants’ investigator-generated identification numbers [14].

### 2.2. Demographic Data

The participants’ age, gender, race, income, marital status, employment status, and education were collected using the National Institutes of Nursing Research demographic form, Bethesda, MD, USA [15]. The participants also completed a list of their medical conditions and medications.

### 2.3. Hair Cortisol Concentration

Approximately 30–50 strands of hair were taken from the posterior area of the scalp. Hair was assayed for the mean cortisol value at the OSU College of Nursing using protocol adapted by D’Anna-Hernandez et al. [16] and Meyer et al. [17]. To prep for assay, the hair samples were cut, then washed twice with isopropanol, and dried over 1 to 3 days. A total of 10–75 mg of hair was placed into a microcentrifuge tube, minced, and then ground in Retsch 400 Mill. A total of 1.1 mL of HPLC-grade methanol was added to the ground sample and incubated for 18–24 h at room temperature with constant agitation. The tubes were centrifuged at 5000× *g* for 5 min at room temperature to pellet the powdered hair. The entire amount (~1 mL) of supernatant was transferred to a clean microcentrifuge tube, and the methanol was removed by evaporation using a stream of air for 6–8 h at room temperature. The cortisol extract was reconstituted in 100 uL of Salimetrics (https://salimetrics.com/wp-content/uploads/2018/03/salivary-cortisol-elisa-kit.pdf, accessed on 2 February 2022) immunoassay cortisol analysis diluent buffer. Samples were assayed in duplicate and intra-assay coefficients of variation calculated. HCC levels are expressed in hair as pg/mg and generally logged due to skewed distributions as needed.

### 2.4. 4-Item Cohen’s Perceived Stress Scale (PSS)

The 4-item subscale of the Cohen’s PSS (Mind Garden, Inc, Newbury Park, CA, USA) is believed to be a more accurate measure of perceived stress in African Americans than the 10-item PSS, with a Cronbach’s alpha of 0.60 [18]. Participants were asked to consider the following questions in relation to the last month: (1) “How often have you felt that you were unable to control the important things in your life?”; (2) “How confident have you felt about your ability to handle your personal problems?”; (3) “How often have you felt that things were going your way?”; and (4) “How often have you felt difficulties were piling up so high that you could not overcome them?” The responses were coded in a Likert-scale format ranging from 0 (never) to 4 (very often).

### 2.5. SF-36 Mental Components Summary

The SF-36 Mental Components Summary (Rand Corporation, Santa Monica, CA, USA) survey assesses limitations in physical and mental health and social role due to health problem, with a Cronbach’s alpha of 0.9 [19]. The 36 questions are a mix of yes/no questions and Likert-scale items ranging from 1–6 responses. They assess limitations in social activities because of physical or emotional problems; limitations in usual role activities because of physical health problems; bodily pain; general mental health; limitations in usual role activities because of emotional problems; and vitality and general health perceptions [19].

### 2.6. Blood Pressure and Mean Arterial Pressure

The diagnosis of hypertension was based upon the participant’ self-report that a provider gave them a diagnosis of hypertension. Additionally, the guidelines for hypertension were a systolic 130–139 or diastolic between 80–89 as per the 2017 American College of Cardiology and American Heart Association [1]. Blood pressure data were obtained using a standardized protocol [20]. Systolic and diastolic blood pressure were collected using an Omron (Omron, Kyoto, Japan) automatic blood pressure machine [21]. Cuff sizes used for collection ranged in size and were individualized to each participant to ensure proper readings. Participants sat, inactive, for five minutes, with legs uncrossed, before collection of blood pressure. Readings were taken three times with a one-minute break in between each collection. The three measurements of systolic and diastolic blood pressure were averaged, and this average was used in the analysis. The average SBP and DBP was used to calculate the MAP (MAP = (SBP + 2 [DBP])/3) [22]. Systolic blood pressure + 2 [DBP]/3 was used to calculate MAP.

### 2.7. Statistical Analysis

Percentages, means, and standard deviation were calculated for each variable. SPSS 27 was used for data analysis [23]. Data were examined for outliers and distribution of data points using (1) histograms with a normal distribution curve line and (2) skewness and kurtosis analysis of the data. Skewed data were adjusted using a log-10 transformation. Pearson correlation was used to determine associations between variables and a two-tailed test at a significance level of 0.05 [24]. Three multiple linear regression models were conducted with (1) SBP, (2) DBP, and (3) MAP as an outcome variables (dependent variables). Mental well-being, perceived stress, and HCC were predictor variables (independent variables) in each model. Only cases with HCC results were included in all of the statistical analyses.

## 3. Results

There were 38 participants from the parent study, of which 81.6% (31) were women. Twenty-five of the thirty-eight total participants provided a hair sample. Among the thirteen, six did not have enough hair, one refused to provide a hair sample collection, and six did not provide a reason. Descriptive characteristics of the 25 participants included in this secondary analysis are represented in Table 1.

Our hypotheses were not supported (Table 2). There was not a positive correlation between HCC and perceived stress, HCC and blood pressure, HCC and mental well-being, or HCC and MAP. A trend towards significance was noted in the relationship between hair cortisol and perceived stress. The relationship between mental well-being and perceived stress and mental well-being and diastolic blood pressure were significant. SBP, DBP, and MAP were significantly correlated.

In the regression model, we investigated whether mental wellbeing, perceived stress, and HCC could predict SBP, DBP, and MAP. The results of the regression indicated that mental well-being was a significant predictor of DBP and MAP only (Table 3). For every unit increase in well-being predicted, there was a 0.42 decrease in DBP and a 1.10 unit decrease in MAP.

## 4. Discussion

HCC provides an easy-to-use method to measure stress. The purpose of this secondary analysis of a mind–body intervention pilot was to examine the relationships between HCC, perceived stress, mental well-being, and cardiovascular health in African American older adults. HCC was not significantly associated with any of the study variables, including MAP. The lack of significant relationship between perceived stress and hair cortisol has been reported by others [24,25,26]. In one study, Lehrer et al. found a direct association between perceived stress and hair cortisol, but once they controlled for low resilience, the association between HCC and perceived stress disappeared [10]. We did not study resilience; future studies focused on African Americans should include this measure and other psychological factors.

We found that mental well-being had an inverse relationship with diastolic blood pressure—lower diastolic blood pressure was associated with higher mental well-being. This also confirms findings of other studies of emotional well-being and blood pressure for the prevention of cardiovascular disease. In line with findings by Konerman et al., as mental well-being increased, diastolic blood pressure decreased [27]. In our study, mental well-being and cardiovascular health (DBP and MAP) were significant, but not for SBP. In one of the largest studies (*N* = 1300), higher emotional well-being were associated with higher SBP [11]. DBP was not reported. However, this study was conducted in Germany and did not include Blacks/African Americans. For clinicians, assessing stress and mental well-being and providing non-pharmacological tools, such as physical activity, healthy eating, and mindfulness meditation, may aid in the reduction of high blood pressure and improve cardiovascular health [13].

One factor that may have yielded null results in this study was the overall health of the participants. Their emotional well-being scores were higher than the average ratings for adults over 65 years old. Participants also had controlled hypertension, reported low levels of stress, and were highly educated. A study found that African Americans reporting higher SF-36 mental and *p*-scores were associated with increased adherence to hypertension management, which may also explain the null results for this study [27].

The limitations related to methodology, specifically the underpowered sample, which limited the statistical analyses, may introduce sampling biases in that the sample may not reflect the African American population of interest. Additionally, the correlation design does not infer probability or causation. Nonetheless, other researchers can use the clear procedural methodology reported here to seek to verify our results with a larger, more ethnically and racially diverse sample and take other psychological factors associated with stress into consideration. A longitudinal design and larger sample would also strengthen a future study. Despite these limitations, this study provided valuable preliminary information on the use of HCC in African American older adults with hypertension and the associations between mental well-being and cardiovascular health.

## 5. Conclusions

Older African Americans are often underrepresented in research. This study contributes to the knowledge of objective physiologic data regarding the relationship between biological and psychological health that may contribute to hypertension, demonstrating that increased mental well-being was associated with decreased perceived stress and diastolic blood pressure in a sample of African American adults 65 years of age and older. Future studies should examine factors that may contribute to the null findings in this study, and this study as well as future evidence should inform the development of interventions that address blood pressure management and mental well-being in African American older adults with hypertension.

## Figures and Tables

**Table 1 geriatrics-07-00053-t001:** Characteristics of the participants ^1^.

Variable	*N* = 25	Range
Age in years	72.2 (5.2)	64–85
Gender (in female) (%)	23 (92.0)	
Race (*n* African American) (%)	25 (100.0)	
Education attainment (some college of above) (%)	15 (60.0)	
Employment (*n* retired) (%)	20 (80.0)	
Marital Status		
(*n* divorced or never married) (%)	15 (60.0)	
(*n* widowed) (%)	7 (28.0)	
(*n* married or domestic partner) (%)	3 (12.0)	
Annual income (median)	1684.7 (804.9)	771–6000
SF-36 Mental Well-being	79.2 (19.7)	20–100
4-Item Perceived Stress Scale	1.7 (0.8)	0–3.3
Hair cortisol concentration	2.2 (1.4)	0.57–5.7
Average systolic blood pressure	136.4 (19.6)	100–171
Average diastolic blood pressure	76.6 (12.3)	55–104
Mean arterial pressure	187.5 (24.7)	137.3–237.0

^1^ Data provided as mean (standard deviation) unless otherwise noted.

**Table 2 geriatrics-07-00053-t002:** Pearson’s correlations for the variables in analysis (*N* = 25).

	Mental Well-Being	Perceived Stress	HCC	SBP	DBP	MAP
Mental Well-Being (SF-36)		**r = −0.497** ***p* = 0.011 ***	r = 0.029 *p* = 0.891	r = −0.211 *p* = 0.312	**r = −0.458** ***p* = 0.021 ***	r = −0.320 *p* = 0.119
4-Item Perceived Stress Scale			r = 0.197 *p* = 0.346	r = 0.039 *p* = 0.855	r = −0.095 *p* = 0.653	r = −0.001 *p* = 0.997
Hair cortisol concentration (HCC)				r = −0.009 *p* = 0.966	r = −0.099 *p* = 0.638	r = −0.040 *p* = 0.849
Average systolic blood pressure (SBP) *					**r = 0.494** ***p* = 0.012***	**r = 0.957** ***p* = 0.000 ***
Average diastolic blood pressure (DBP) *						**r = 0.724** ***p* = 0.000 ***
Mean arterial pressure (MAP)						-

* *p* < 0.05; r = correlation coefficient.

**Table 3 geriatrics-07-00053-t003:** Multiple Linear Regression Models for SBP, DBP, and MAT.

Outcome	Sample Size	Predictor	Predictor Slope	Lower CL	Upper CL	Predictor *p*-Value	R Square
SBP	25	INTERCEPT	160.20	104.95	215.45		0.05
	.	Mental Well−Being	−0.26	−0.77	0.26	0.31	
	.	Perceived Stress	−2.40	−16.02	11.22	0.72	
	.	HCC	0.24	−6.30	6.78	0.94	
DBP	25	INTERCEPT	121.39	92.63	150.14		0.35
	.	Mental Well−Being	**−0.42**	**−0.69**	**−0.15**	**<0.01 ***	
	.	Perceived Stress	−6.99	−14.08	0.09	0.053	
	.	HCC	0.04	−3.36	3.45	0.98	
MAP	25	INTERCEPT	402.97	306.34	499.61		0.24
	.	Mental Well−Being	**−1.10**	**−1.99**	**−0.20**	**0.02 ***	
	.	Perceived stress	−16.39	−40.21	7.43	0.17	
	.	HCC	0.33	−11.11	11.77	0.95	

* *p* < 0.05; confidence level (CL) 95%; hair cortisol concentration (HCC); systolic blood pressure (SBP); diastolic blood pressure (DBP); and mean arterial pressure (MAP).

## Data Availability

Data supporting this study are housed at The Ohio State University College of Nursing. De-identified data can be accessed by contacting the corresponding author at wright.2104@osu.edu.

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
