# Peer review of "Hair Cortisol Concentration, Perceived Stress, Mental Well-Being, and Cardiovascular Health in African American Older Adults: A Pilot Study"

_geriatrics, 2022, doi:10.3390/geriatrics7030053_

Round 1
Reviewer 1 Report
Thank you for allowing me the privilege to review this brief report title "Hair Cortisol Concentrations, Perceived Stress, Blood Pressure, and Mental Health Well-Being in African American Older Adults: A Pilot Study." The content in this report does provide content of interest within an population at-risk for poor health outcomes, specifically hypertension related to stress. Additionally, this report provides a novel, objective approach to measuring stress using hair cortisol levels. Despite the limitations related to methodology- specifically the underpowered sample limiting statistical analyses while potentially introducing sampling biases in which the sample may not reflect the African American population of interest, the brief does provide clear procedural methodology that is informative to other scientists in terms of replication. Moreover, the background/rationale for conducting the study based on proposed hypotheses were well developed. Below are my specific editorial suggestions aimed at improving the overall scientific impact/interest to readers:
Introduction (Line 75): Since this study is a secondary analysis, it may not be appropriate to consider brief hypotheses a priori in nature unless the expressed hypotheses were primary aims of the DASH study.
Methods: Study Design and Participants:
(1) There is no reference of when this data was collected.
(2) Although reference is made to the design (randomized control trial) of the parent study, the sample size collected within the parent study is not presented. Thus, was a specific sub-set of study participants used within the secondary analysis or was the entire sample from the parent study used? If a sub-set was used, what was the inclusion/exclusion criteria associated with selecting this sub-set of individuals?
Table 2: Bivariate Correlations: this table difficult to read due to formatting issues.
Author Response
Thank you for your review.

Reviewer 2 Report
1- General comment
I want to congratulate the authors for the effort and time spent conducting this pilot study. In general, the study is very pertinent and well designed. The results demonstrated significant inverse relationships between mental well-being and perceived stress, and mental well-being and diastolic blood pressure in African American older adults with hypertension. Therefore, these results indicate the importance of implementing effective intervention strategies to manage stress levels in older African Americans with hypertension. Although I consider that the article has potential for publication, there is space for improvement. Specifically, I recommend that the authors perform a multiple linear regression analysis using mean arterial pressure as the dependent variable and mental well-being, perceived stress, and hair cortisol concentration as the independent variables. In addition, other aspects need further explanation. Thus, I recommend major revisions. Below, I provide some suggestions.
2- Abstract
The authors should rewrite the abstract with the regression analysis results.
LL38: Please, consider including more relevant keywords (you can insert 3 to 10) to increase the visibility of your study.
3- Introduction
The rationale of the study is well-presented during the introduction. Nevertheless, I suggest that the authors emphasize the research problem before presenting the study aim. What are the research gaps identified in the literature that need further explanation? What novel aspect will your study bring to the scientific community? I recommend that the authors write an additional paragraph before the study aims to highlight these aspects.
LL54-55: “…African Americans report higher stress levels and have greater exposure to chronic stressors in their lifetime than whites…”. I recommend that the authors write an additional sentence with possible causes for these differences with the respective study reference.
LL63: Please, consider rewriting the sentence to the following: “However, there is limited research on hair cortisol concentration and hypertension among African Americans”.
4- Materials and Methods
Did you collect data regarding height and body mass? It would be interesting to report these data since some literature indicates an inverse relationship between height and blood pressure (https://doi.org/10.1159/000514205; https://doi.org/10.1186/s40885-021-00164-4).
LL145: What blood pressure guidelines did you use to categorize the participants as hypertensive? Please, clarify this issue and add the reference values and the study reference. In addition, I recommend that the authors calculate the mean arterial pressure (MAP = (SBP + 2 [DBP]) / 3). This study can help you https://doi.org/10.1161/JAHA.117.005477.
LL155: I suggest that the authors perform a multiple linear regression analysis using mean arterial pressure as the dependent variable and mental well-being, perceived stress, and hair cortisol concentration as the independent variables. If you find it pertinent, you can also run two separate multiple linear regression analyses using DBP and SBP as the dependent variables and add other outcomes to test their significance in the model. In addition, age and gender should be used as adjustment variables. Finally, check the correlation between mean arterial pressure and the remaining variables.
5- Results
Table 1 – Add the mean arterial pressure results (mean ± SD; range)
Table 2 – Add the correlation results between the mean arterial pressure and the remaining variables.
LL176-178: Please, consider removing the results in parenthesis because they are already reported in Table 2. Try not to repeat results.
6- Discussion and Conlusions
These sections should be reformulated based on the regression analysis results and the correlation between mean arterial pressure with other variables.
Author Response
Thank you for your review.

Reviewer 3 Report
Ericka L. Richards et al submitted a manuscript titled: “Hair Cortisol Concentration, Perceived Stress, Blood Pressure, and Mental Well-being in African American Older Adults: A Pilot Study”. While the topic could be of interest, the manuscript itself has many ambiguities and flaws.
- It is already well-established that “Chronic stress and hypertension are associated with cardiovascular disease and both are highly prevalent in African Americans”. Hence, more information is needed regarding the background of your study in the Abstract.
- Your sample size is not 38 since only 25 participants provided the hair sample.
- Moreover, 25 participants are such a small sample size that no meaningful statistical analysis can be conducted on it.
- Additionally, it is unclear when was the hair sample taken? Even though the hair cortisol level is an indicator of chronic stress it is still relevant was it taken in a short period around the completion of the mental well-being instruments or maybe in a longer period.
- 4-item perceived stress scale is an ordinal variable and as such should be presented as median (IQR). Moreover, since the scale range is so small it is statistically inappropriate to estimate correlations with it (especially not the Pearson’s correlation!).
- You should add which statistical method was used to estimate the normality of distribution.
Due to the small sample size (even though it is presented as a pilot study), I advise you to collect your full sample size data, revise these several issues and then resubmit your study. Furthermore, in my opinion pilot studies (for this type of studies) should be used for sample size estimations or for conducting some preliminary analyses for the direction of the future study but they do not have the quality or reliability of an original article.
Author Response
Thank you for the review.

Round 2
Reviewer 2 Report
Dear authors,
I read the revised manuscript, and I want to congratulate you on the excellent work in revising the manuscript as requested. In my opinion, the manuscript is ready for publication.
Best regards.